# Application of 3D-Printed, PLGA-Based Scaffolds in Bone Tissue Engineering

**DOI:** 10.3390/ijms23105831

**Published:** 2022-05-23

**Authors:** Fengbo Sun, Xiaodan Sun, Hetong Wang, Chunxu Li, Yu Zhao, Jingjing Tian, Yuanhua Lin

**Affiliations:** 1State Key Laboratory of Advanced Ceramics and Fine Processing, School of Materials Science and Engineering, Tsinghua University, Beijing 100084, China; sunxiaodan@tsinghua.edu.cn (X.S.); wht19@mails.tsinghua.edu.cn (H.W.); 2Department of Orthopaedics, Peking Union Medical College Hospital, Chinese Academy of Medical Sciences, Beijing 100730, China; licunxupumch@163.com (C.L.); zhaoyupumch@163.com (Y.Z.); tianjing311@163.com (J.T.)

**Keywords:** PLGA, 3D-printing, bone tissue engineering, biological nanotechnology

## Abstract

Polylactic acid–glycolic acid (PLGA) has been widely used in bone tissue engineering due to its favorable biocompatibility and adjustable biodegradation. 3D printing technology can prepare scaffolds with rich structure and function, and is one of the best methods to obtain scaffolds for bone tissue repair. This review systematically summarizes the research progress of 3D-printed, PLGA-based scaffolds. The properties of the modified components of scaffolds are introduced in detail. The influence of structure and printing method change in printing process is analyzed. The advantages and disadvantages of their applications are illustrated by several examples. Finally, we briefly discuss the limitations and future development direction of current 3D-printed, PLGA-based materials for bone tissue repair.

## 1. Introduction

Treatment of bone defects and their associated disorders remains a real clinical challenge. Bone defects are usually caused by severe external injuries and various diseases in the human body. Bone replacement materials are used when the defect does not heal on its own. Common bone graft materials include autogenous bone, allogeneic bone, and xenogeneic bone. Excellent bone graft materials should meet the following characteristics: (1) have bone conduction matrix—it provides a matrix for cell migration, adhesion, and proliferation and provides an environment for blood vessel growth and new bone formation; (2) bone inducing factor—it can stimulate the differentiation of original and undifferentiated stem cells into chondroblasts or osteoblasts to form new bone; (3) stable biomechanical environment—with appropriate mechanical properties, it can transfer natural biomechanical stress to the surrounding normal bone tissue. If the strength of bone graft material is too low, the whole structure will be destroyed, while if the strength is too high, stress shielding will occur, leading to the surrounding osteoporosis. Autologous bone grafts are derived from the patient itself and basically meet the abovementioned conditions. It has been regarded as the gold standard in the field of bone transplantation and is the first choice for bone graft surgery due to its excellent bone conductivity and bone inductance. However, autologous bone grafts require additional surgical incisions, produce limited amounts of bone, and increase the chance of bleeding and infection. Allogeneic bone is relatively easy to obtain, but the donor is limited, and there is the possibility of various infectious diseases and cross-infection. Xenogeneic bone is usually made of animal bone. It is rich in sources and easy to store. However, there is immune rejection and lack of osteogenic growth factors. Therefore, synthetic bone substitutes have attracted more and more attention as an optimal replacement strategy [1,2,3].

Natural bone tissue has a distinct hierarchical structure, which is gradually formed by smaller structural units. The regeneration of bone tissue goes through a complex biological process and is affected by the surrounding environment [4]. Considering the complex interaction between host tissue cells and graft, a series of biocompatible materials such as inorganic minerals, metal alloys, and biopolymers are used to prepare graft bone [5]. After continuous research, artificial bone graft materials have also experienced continuous development and improvement. Among them, the first generation of biomaterials are inert materials, represented by metals, some synthetic polymers, and biological ceramics. The second generation of biomaterials are biodegradable materials represented by biodegradable polymers, phosphates, and silicates. The third generation of biomaterials are bioactive materials loaded with cells, genes, and growth factors that have the bone-forming ability similar to that of autologous bone. However, bone substitutes remain problematic. For example, lack of matching mechanical properties of bone tissue, rapid repair of bone defects of different sizes, and matching degradation time with osteogenesis time. In addition, safe and effective osseointegration and vascular formation can be achieved for a long time after implantation [6,7].

From a material point of view, properties are directly dependent on composition, structure, and processing. In order to obtain perfect artificial bone graft materials, we need to select more appropriate materials and preparation methods. Polylactic acid–glycolic acid (PLGA), as a representative biodegradable polymer, has attracted extensive attention of researchers. PLGA-based devices have been widely used in the biomedical field [8]. PLGA has good biocompatibility, adjustable degradation, and mechanical properties, as well as applications in various machining properties. PLGA-based materials prepared by different methods have been developed and applied to bone tissue repair in different parts of the body [9,10]. In previous studies, scaffolds, fibers, hydrogels, microspheres, and other applied PLGA materials have been summarized to a certain extent. These application forms are prepared by 3D printing, electrospinning [11,12,13], and other methods [8,14,15]. Compared with other methods, 3D printing technology has been increasingly widely used in tissue engineering and regenerative medicine due to its characteristics of rapid prototyping, structural design, and personalized customization. It can quickly meet the different needs of different shapes and sizes, and the prepared porous structure can meet the biocompatibility required for bone tissue repair [16,17]. This review summarizes the previous work, aiming at the application of PLGA-based scaffolds based on 3D printing technology in bone tissue engineering. The design and preparation of different structures and components of 3D-printed, PLGA-based scaffolds, in vitro and in vivo performance evaluation of scaffolds, and the latest research results in recent years were further introduced. Finally, we summarize the challenges and future development direction of ARTIFICIAL bone grafting based on PLGA.

## 2. Physical Properties and Degradation Behavior of PLGA

PLGA is a biodegradable functional polymer organic compound synthesized by the polymerization of two monomers, lactic acid (LA) and glycolic acid (GA), it has good biocompatibility and no biotoxicity [18]. The molecular weight (g/mol) of PLGA varies from thousands to hundreds of thousands in different LA:GA ratios. In order to meet the processing mode of 3D printing, organic solvents are generally used to dissolve PLGA to prepare printing ink, such as acetone and 1,4-dioxane. As an artificial bone graft, its biodegradability is directly related to bone formation, which is of great significance. For PLGA, LA:GA ratio, molecular weight, and end-group of molecular chain all affect the degradation rate of PLGA scaffolds [19,20,21]. The main degradation modes of PLGA are hydrolysis and ester bond autocatalytic degradation. Van der Waals forces and hydrogen bonds are broken when water penetrates into the amorphous region of the polymer matrix. The continuous breakage of the main chain covalent bond leads to the decrease in molecular weight and chain integrity, the continuous decrease in mass, the destruction of the integrity of the scaffold, and the final dissolution in the surrounding medium [18,22]. The results showed that the degradation rate of PLGA decreased with the increase in LA:GA ratio. In addition, LA and GA as degradation products are acidic. They will make the local microenvironment a weakly acidic environment, which may produce an inflammatory response [23,24].

## 3. Regulation of Compositions in 3D-Printed, PLGA-Based Scaffolds

Combined with clinical imaging techniques such as microcomputed tomography (Micro-CT) and magnetic resonance imaging (MRI), 3D printing can easily reconstruct the required 3D model and accurately generate bone grafts that match the defect area. The limitation of current research mainly lies in the lack of sufficient mechanical strength due to the influence of porous structure on 3D-printed scaffolds. Moreover, the scaffold degradation rate did not match the time of osteogenesis. Although 3D-printed PLGA scaffolds alone cannot meet the requirements of bone grafts for osteogenic performance and mechanical strength, they still play an important role in the research and application of 3D-printed bone grafts because of their ability to accurately regulate the degradation rate [25].

Studies have shown that PLGA scaffolds with high LA:GA ratio and ester-terminated groups have higher mechanical strength and degradation time. In the 3D printing process, the selection of LA:GA ratio is particularly important and will directly affect the performance of the final stent. The actual degradation process of scaffolds is in a complex biological environment, which is affected by many factors [26,27]. Ma, C.H. et al. reported the degradation of 3D-printed PLGA scaffolds in three conditions, which are phosphate buffered saline (PBS) solution, microchannel, and shaking incubator. The solution was circulated under microchannel conditions. Measured parameters of scaffolds include mass loss, water absorption, structural changes, and porosity changes. The results showed that stress and flow perfusion factors were two main factors affecting the degradation behavior of 3D-printed PLGA scaffolds [28]. In addition to the physicochemical properties of the original PLGA scaffold, its biological properties were also evaluated using a rabbit model [29]. The results showed that the PLGA scaffolds had good porous structure, biocompatibility, and bone conductivity. Since artificial bone implants need to have the mechanical strength similar to human bone and strong osteogenic ability, PLGA alone is obviously unable to meet these requirements. However, it also provides guidance for the modification direction of 3D-printed scaffolds. The ideal bone repair scaffold should have appropriate printing performance, mechanical strength, degradation rate, osteogenic properties, and other characteristics; so, it is necessary to optimize and improve the PLGA scaffold [30].

### 3.1. Organic/Inorganic Composite Scaffolds

Based on the above requirements, it is an excellent alternative strategy to prepare composite scaffolds by combining PLGA with inorganic materials with bone transduction and bone induction functions. Among them, the most commonly used are hydroxyapatite (HA) and tricalcium phosphate (TCP), which are natural components in bone tissue and have strong bone conductivity and bone induction [31,32,33]. The addition of TCP can also improve the acidic environment caused by PLGA degradation and improve the compressive strength and elastic modulus of PLGA scaffolds. Hwang developed a microsized polycaprolactone (PCL)/PLGA/β-TCP granular bone graft that blends collagen matrix to create a composite block. These composites have excellent plasticity and can be cut or compacted into the desired shape. In addition, the collagen matrix between the bone grafts prevents them from improving while avoiding structural destabilization, as shown in Figure 1 [34].

Although organic–inorganic composite materials have improved the osteogenic performance of PLGA scaffold, they have not solved the problem of insufficient mechanical strength. Lai prepared a novel porous PLGA/TCP/Mg (PTM) scaffold. The scaffolds have good mechanical properties, corrosion resistance, and biological activity. This study systematically analyzed the physical properties of PTM. The osteogenic performance and biosafety of PTM scaffolds were evaluated by using *rabbit* femur bone defect model. Compared with PLGA/TCP (PT) scaffolds, the compressive strength of PTM scaffolds increased from 1.5 ± 0.1 MPa to 3.7 ± 0.2 MPa, and Young’s modulus increased from 45.7 ± 5.4 MPa to 114.9 ± 15.4 MPa. The *rabbit* model of femur defect also demonstrated the excellent osteogenic performance of PTM scaffolds, as shown in Figure 2 [35]. Yang et al. prepared a PLGA/HA scaffold for large-size bone defect repair, not only for small-size bone defect repair. The results showed that the composite scaffolds had good osteogenic ability and antibacterial activity in treating large bone defects. The above work indicates that 3D-printed, PLGA-based scaffolds have great potential in repairing bone defects of different types and sizes [36].

### 3.2. Growth Factor or Drug Functionalization in Scaffolds

There are limits to using materials alone to improve biocompatibility. To further improve the biocompatibility of 3D-printed scaffolds, growth factors or drugs can be loaded onto the scaffolds. They generally have strong osteogenic ability and can effectively improve the lack of biocompatibility of scaffolds. Growth factors represented by bone morphogenetic protein 2 (BMP-2) and bone-promoting drugs have been extensively studied in the field of bone tissue engineering [37,38]. Shim et al. prepared a 3D-printed PCL/PLGA scaffold loaded with BMP-2, and studied the different modes of transportation. In this paper, collagen and gelatin were used to encapsulate BMP-2, respectively. The effects of long-term and short-term delivery were observed [39]. The results showed that long-term BMP-2 delivery could upregulate the expression of osteogenic genes at the same dose. They also showed that stents that delivered BMP-2 over a longer period of time had better osteogenesis in the body than those that delivered BMP-2 over a shorter period of time. However, the rapid release of BMP-2 induces an inflammatory response in the short term, suggesting that controlling BMP-2 release per unit of time is particularly important. Therefore, they further printed PCL/PLGA/β-TCP scaffold with slow release of BMP-2 for *rabbit* skull defect model repair [40]. Release curve results showed that BMP-2 could be released continuously in the composite scaffold for 28 days. Compared with the scaffolds without BMP-2 loading, the composite scaffolds had better osteogenic ability. Deng et al. loaded BMP-2 into PLGA/nHA scaffolders and used chitosan (CS) as the nanocarrier [41]. The cumulative release of PLGA/nHA/CS/BMP-2 scaffold was only 9.54 ± 0.86% in 48 h and 61.38 ± 2.39% in 30 days, reaching the expected sustained release effect. The results showed that PLGA/nHA/CS/BMP-2 scaffold successfully repaired the bone defect area. The above work has proved that loading growth factor is a good modification method for 3D-printed scaffolds [42,43].

More effective, less toxic, more stable, and less expensive bone implants are the development direction; although, BMP-2 has strong osteogenic activity. However, when used in vivo as a biosynthetic bioactive agent, it has a number of inherent limitations, including a short half-life, low activity, side effects beyond physical doses, and potential immune responses during long-term use. Therefore, it is of great significance for bone tissue regeneration engineering to find reliable growth factor substitutes. Lin et al. developed PLGA/β-TCP stents loaded with salvianolic acid B for spinal fusion therapy, as shown in Figure 3 [44]. They evaluated the bone fusion ability of stents in a *rat* spinal fusion model and showed that salvianolic acid B had a favorable effect on promoting mineral deposition, bone formation, and angiogenesis. Another study reported that 3D-printed PLGA/TCP composite scaffold combined with the bioactive plant molecule icariin can also promote ulna regeneration in rabbits [45]. The results also showed that icariin promoted bone formation and angiogenesis in a dose-dependent manner, as shown in Figure 4.

A large number of 3D-printed, PLGA-based scaffolds have been studied based on different components. We found that the newly formed bone tissue can only migrate slowly from the periphery of the scaffold to the center due to limited oxygen and nutrient exchange, uneven cell distribution, and migration, especially in animals with large bone defects. Therefore, 3D-printed scaffolds need to be capable of internal vascularization [46,47,48]. The regeneration and remodeling of bone tissue require the synergistic effect of bone induction and vascularization. However, due to different mechanisms, it is difficult to achieve these two biological functions simultaneously using only one bioactive growth factor. Similar to BMP-2, vascular endothelial growth factor (VEGF) is the most commonly used growth factor in bone tissue engineering in the past [49]. It is worth noting that growth factors such as BMP-2 and VEGF need to ensure their activity during use. This has greatly limited the preparation method of stents, and their dosages and potential risks in humans are still worthy of further study. Cheng et al. found that cucurbitacin B (CuB), a tetracyclic terpene derived from Cucurbitaceae family plants, was beneficial in inducing angiogenesis. PLGA/β-TCP/Cub composite scaffolds were prepared; their angiogenesis and osteogenesis were verified by in vitro experiments and the *rat* skull defect model. The results showed that CuB stimulated angiogenesis by upregulation of VEGF signaling pathways. PT/CuB stent significantly promoted neovascularization and bone regeneration in the rat model of critical size skull defect compared with that without CuB scaffolds, as shown in Figure 5 [50].

### 3.3. Composite Scaffolds Loaded with the Cells

In recent years, in addition to growth factors and drugs, stem cells (BMSC, ADSCs, etc.) also have a promising application prospect in bone tissue engineering [51]. Cells are implanted into the scaffold material to form stent-based cell implants that mimic the microscopic structure of biological tissues. It should be noted that cells are subjected to continuous mechanical force during the printing process, which may lead to cell damage, resulting in unsatisfactory cell numbers and distribution in the scaffolds. More recently, Probst et al. prepared a TCP/PLGA scaffold with ADSCs. The potential of TCP/PLGA scaffolds to repair bone defects with or without cells was evaluated through a miniature *pig* mandibular defect model [31]. Micro-CT results showed that TCP/PLGA scaffolds with ADSCs had higher new bone volume and bone mineral density. The research brings 3D-printed scaffolds loaded with living cells closer to clinical use.

Table 1 summarizes the regulatory characteristics of PLGA-based stents. Current studies confirm that PLGA-based 3D-printed scaffolds have the potential to be used as artificial bone repair materials. A series of in vitro and in vivo experiments have proved that PLGA-based composite scaffolds have excellent angiogenesis and osteogenesis performance. However, there are still many questions to be answered before clinical application. In the printing process, the printability of materials, the accuracy of printing scaffolders, the preparation speed after mass production, and the maintenance of biological components’ activity are still facing great challenges. In addition, the printing method of the current bracket is relatively simple. Changes such as pore size, porosity, and hierarchical structure changes have been shown to be effective [52,53].

## 4. Structure Design of 3D-Printed, PLGA-Based Scaffolds

In order to store the specified volume of material continuously and accurately, the flow and speed of the dispensing head must be carefully controlled; otherwise, the material will accumulate or break. Due to the viscoelasticity of the polymer material, the scaffold structure may be deformed to some extent during the deposition, resulting in the deviation of the aperture structure. Therefore, understanding the mechanical properties and process parameters of scaffolds is an important process that affects the pore size and porosity.

In addition to preparing single-structure scaffolds, multilayer scaffolds can also be designed and modified with different functions to meet different requirements. Jia et al. developed a multilayered scaffold (MLS) that successfully mimics the different spatial structures and natural complex components of osteochondral tissues. The three layers of the scaffold correspond to hyaline cartilage, calcified cartilage, and subchondral bone of natural tissue, respectively. They are designed to support the proliferation and differentiation of endogenous mesenchymal stem cells to repair multiple tissue defects in the osteochondral region. The biomimetic MLS repair effect was evaluated in *goat* model. After 48 weeks postoperatively, the upper cartilage of MLS group was hyaline and the subchondral bone was healthy. In addition, biomimetic MLS also significantly improved the biomechanical and biochemical properties of new osteochondral tissue. These results demonstrate the potential of MLS as a repair for osteochondral defect, as shown in Figure 6 [54].

Typically, each layer is rotated 90° to produce a square aperture; so, it has been considered whether the performance of the scaffold could be further improved through a specific aperture. Kim et al. investigated the effect of composite materials with specific pore structure on the reaction in *rabbit* femur defect model and modified PLGA/β-TCP scaffold with HA coating. The results showed that the scaffold was biocompatible and biodegradable for up to 12 weeks. HA coating could promote the healing of rabbit femur injury [55]. It is expected that the specific pore structure will contribute to the increase in endogenous protein adsorption and cell attachment, and ultimately promote bone regeneration. However, the results showed that the effect of the specific structure was minimal (much less than 10% of the new bone area).

## 5. Printing Methods of 3D-Printed, PLGA-Based Scaffolds

To address the limitations of traditional scaffold manufacturing techniques, 3D printing is used to create complex scaffolds with customized external shapes and repeatable internal structures. These techniques can create a network of blood vessels in the scaffold, which facilitates the transfer of nutrients to the scaffold and improves the survival rate of biological tissue after implantation. So far, various 3D technologies have been successfully applied to tissue engineering and received good reviews [56,57]. In these technologies, extrusion conveying semifused polymers or printing inks are generally adopted [58]. Extrusion printing was inspired by printer technology [59], and most of the devices used in tissue engineering applications are adapted from commercial printers. This method has been successfully used to create 3D scaffolds and print living cells [60].

Low-temperature 3D printing can prepare scaffolds with excellent biological properties, which can simulate extracellular matrix; provide nutrition, oxygen, and growth factors; promote cell proliferation; and support three-dimensional structure of tissues. Scaffolds with abundant pore structure play an important role in tissue engineering, and the development of their preparation technology is very important. Compared with traditional manufacturing technology, low-temperature 3D printing technology can accurately control the pore size, interpore connectivity, porosity, and pore spatial distribution of the scaffold by adjusting the solvent content [35].

In the present study, the possibility of producing bone substitutes from 3D-printed, PLGA-based composites using fused deposition modeling (FDM) technology was evaluated. First, the composite materials produced do not require the use of any chemical solvents and avoid the limitations of using solvents. Secondly, compared with low-temperature 3D printing technology, scaffolds prepared by FDM reduce microporous structures and have higher mechanical strength, which is more suitable for bone repair sites requiring higher mechanical properties, such as spine and knee joint [61]. Carlier et al. was the first to use FDM to study a monoclonal antibody (mAb)-loaded device. PLGA and mAb powder (15% *w*/*w*) were prepared into printable filamentous material. The FDM process was optimized to print the filaments without changing the stability of monoclonal antibodies. All the excipients have good stability in FDM process. The sustained release profile displayed by 3D printing equipment has a low burst effect. The binding capacity of the mAb remained at 70% throughout the preparation process. The results show that FDM can be used to produce mAb-loaded devices with good stability, affinity, and slow-release characteristics, as shown in Figure 7 [62]. In general, higher porosity results in good biocompatibility. However, with the increase in porosity, the mechanical strength of the scaffold will decrease. In practical use, the needs of the receptor site need to be considered. Selecting a more suitable preparation method can guarantee the mechanical strength and improve the osteogenic properties of the material.

## 6. Challenges and Future Perspective

In the past, the development of bone tissue engineering biomaterials based on PLGA has been very rapid. However, there are some limitations that need to be addressed. Currently, all experimental methods are still in the basic stage. In order to successfully develop scaffolds for bone regeneration, we need to further study the specific process of new bone formation, which will help us develop materials with better performance; realize clinical application; and finally, make products industrialized. PLGA has been widely used in the repair of bone defects of different types and sizes due to its precise degradation rate. Although scaffold degradation in the early stage can provide space for osteogenesis, rapid degradation will lead to the destruction of scaffold structure, which is not conducive to long-term bone reconstruction [63,64]. However, most of the current degradation studies remain at the stage of no external force. We know that after the actual bone replacement is implanted, it will be continuously affected by external force, which will greatly affect the degradation rate of the scaffold. Therefore, considering the above factors, it is still a difficult task to design and prepare artificial bone substitutes that can accurately regulate the degradation behavior of bone tissue [65].

Different from natural polymers, PLGA has relatively few ionic molecular groups, leading to unsatisfactory mineralization, which is also one of the reasons for the poor osteogenic performance of PLGA alone. Therefore, PLGA is usually biomineralized and osteogenic using inorganic minerals or biological inducers. However, because the polymer is dissolved in organic solvents or after melting, 3D printing inks have high viscoelasticity. When the inorganic particles are mixed with the printing ink, there will be some agglomeration phenomenon, and the inorganic particles cannot be well-dispersed, which will lead to the uneven mechanical properties of the whole scaffold. However, inactivation and the risk of adverse reactions and ectopic osteogenesis after implantation should not be ignored during stent preparation [66,67]. At present, the application of growth factors in bone regeneration is limited due to its high cost, limited stability due to preparation methods, and potential risks. Therefore, it will be the development trend of practical application in the future to form PLGA composite materials with inorganic components, drugs, and other components with low cost and good stability.

Currently, bone regeneration strategies loaded with growth factors or cells have been extensively studied. However, physical stimuli without cellular or biochemical transmission are also increasing, promising endogenous healing [68,69,70,71]. Since bone graft directly interacts with tissue fluid and cells after implantation, the overall structure of the scaffold will greatly affect the final bone repair effect. By studying the natural bone tissue structure and preparing multilayer scaffold structure, it is expected to realize the synergistic effect of different functions in the achievement process. Previous studies have promoted the interaction between artificial bone graft and cells and the integration with host bone tissue by regulating the overall topological structure of the scaffold [72]. From the microscopic perspective, the surface morphology of scaffold can also regulate cell fate and mediate bone regeneration [73,74]. Therefore, we should not be limited to the existing scaffold preparation technology, and the development of PLGA scaffolds with unique structure and surface morphology may be one of the future development directions. The other direction of development is to prepare multiple forms of composite scaffolds by various technologies to make up for the shortcomings of existing scaffolds.

## 7. Conclusions

As a key material for bone tissue engineering applications, PLGA has made remarkable progress in the past. Various studies have shown that PLGA-based materials have various advantages in the field of bone tissue repair, especially in the precise control of degradation rate. Loading inorganic particles and drugs into PLGA matrix can effectively improve the biocompatibility of scaffolds and avoid the limitation of using traditional growth factors. Although there is still a long way to go for clinical application, we need to take into account the various complex mechanisms involved in the process of new bone formation and improve the existing detection and preparation methods, such as the stability and dynamic degradation curve of scaffolds under the simultaneous action of bone cells and external forces. We need to overcome the limitations of existing 3D printing technologies and further understand the characteristics of various scaffolds. Different scaffold preparation methods and multilayer structure design have shown excellent osteogenic properties. Therefore, more comprehensive structures and functions of artificial bone repair materials are needed in the future, including the design of topological bone substitutes and the further development of hybrid scaffolds combined with a variety of biological manufacturing technologies.

## Figures and Tables

**Figure 1 ijms-23-05831-f001:**
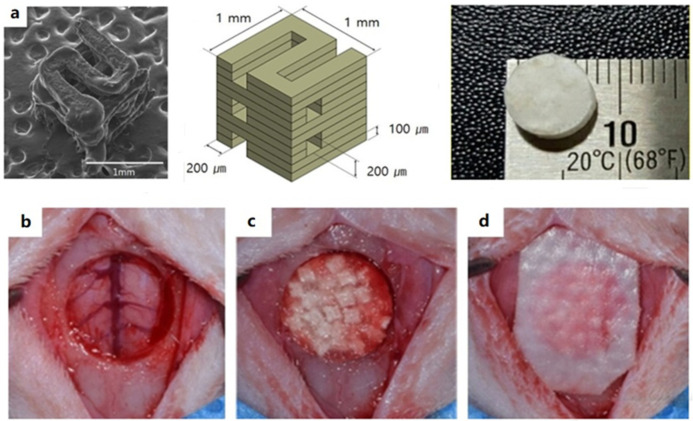
Bone defect repair model based on PCL/PLGA/β-TCP scaffolds: (**a**) SEM image and PCL/PLGA/β-TCP scaffold simulation structure; (**b**) porous bone defect structure (8 mm-diameter); (**c**) defect area filled with PCL/PLGA/β-TCP composite bone graft; (**d**) defect area filled with collagen membrane, Reprinted with permission from ref. [34]. Copyright 2017 MDPI.

**Figure 2 ijms-23-05831-f002:**
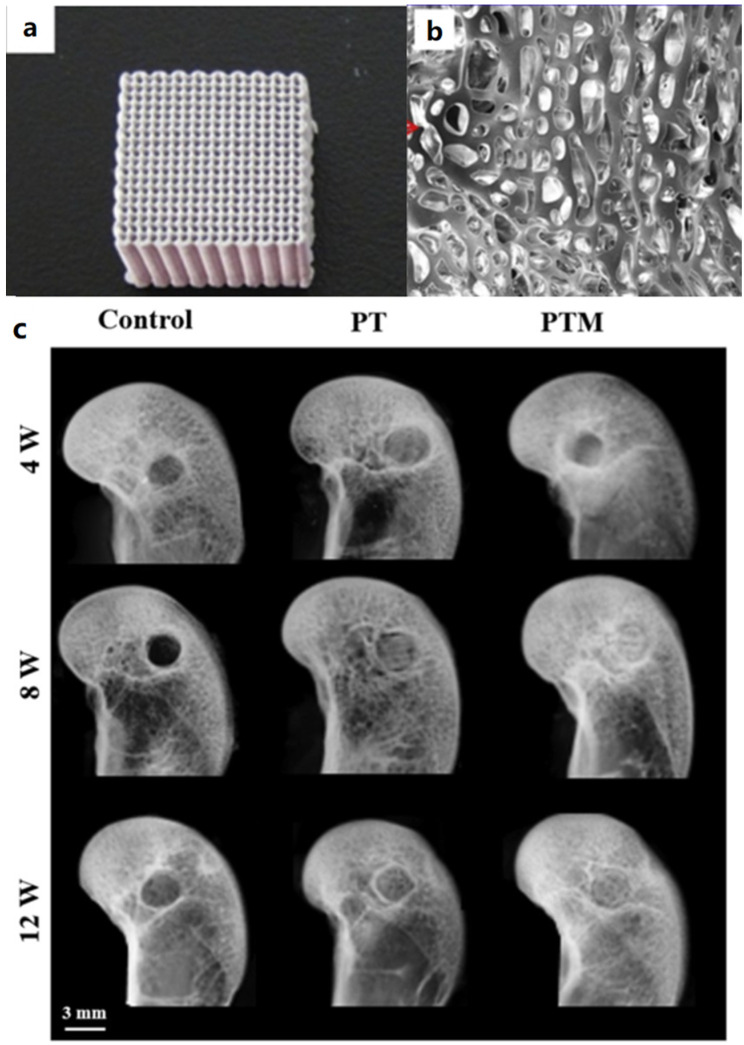
PTM scaffolds show excellent mechanical strength and osteogenic properties. (**a**) PTM scaffolds. (**b**) The porous surface structure ensures good biocompatibility of PTM scaffolds. (**c**) *Rabbit* model of femur defect repaired with PTM scaffolds, Reprinted with permission from ref. [35]. Copyright 2019 Elsevier.

**Figure 3 ijms-23-05831-f003:**
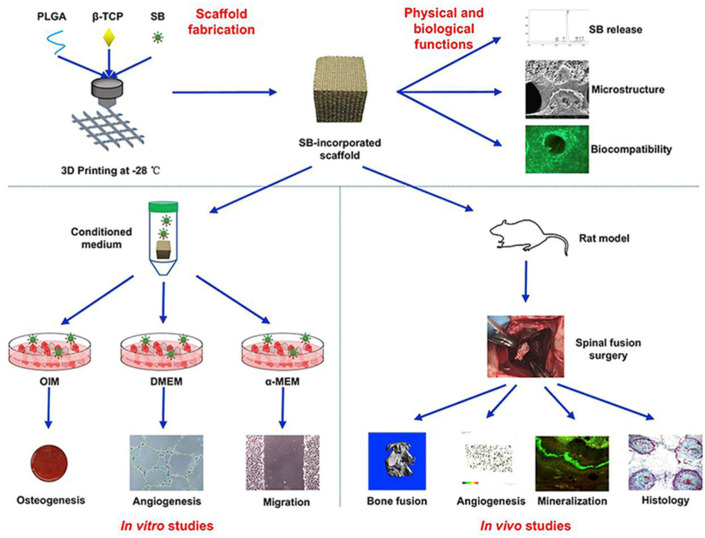
PLGA/β-TCP composite scaffold incorporating salvianolic acid B promotes bone fusion by angiogenesis and osteogenesis in a *rat* spinal fusion model, Reprinted with permission from ref. [44]. Copyright 2019 Elsevier.

**Figure 4 ijms-23-05831-f004:**
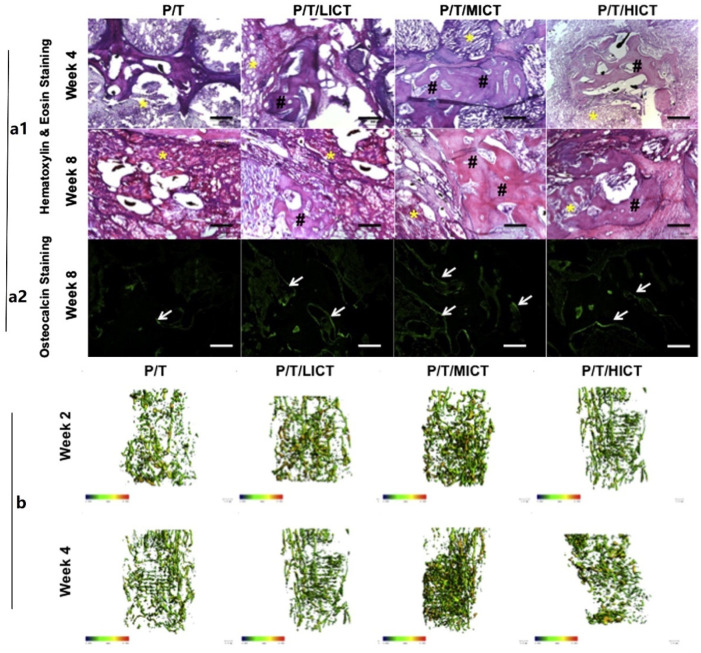
Icariin can effectively improve osteogenesis and angiogenesis. (**a1**) Sagittal section of decalcified bone defect with H&E staining (yellow *, scaffold; black #, new bone; black or white bar = 200 μm), (**a2**) Representative immunohistochemistry of osteocalcin expression in osteoblasts and mineralized sites (**b**) Representative micro-CT-based microangiography of vessels formed within the ulnar segmental defect region at weeks 2 and 4 after implantation, Reprinted with permission from ref. [45]. Copyright 2013 Elsevier.

**Figure 5 ijms-23-05831-f005:**
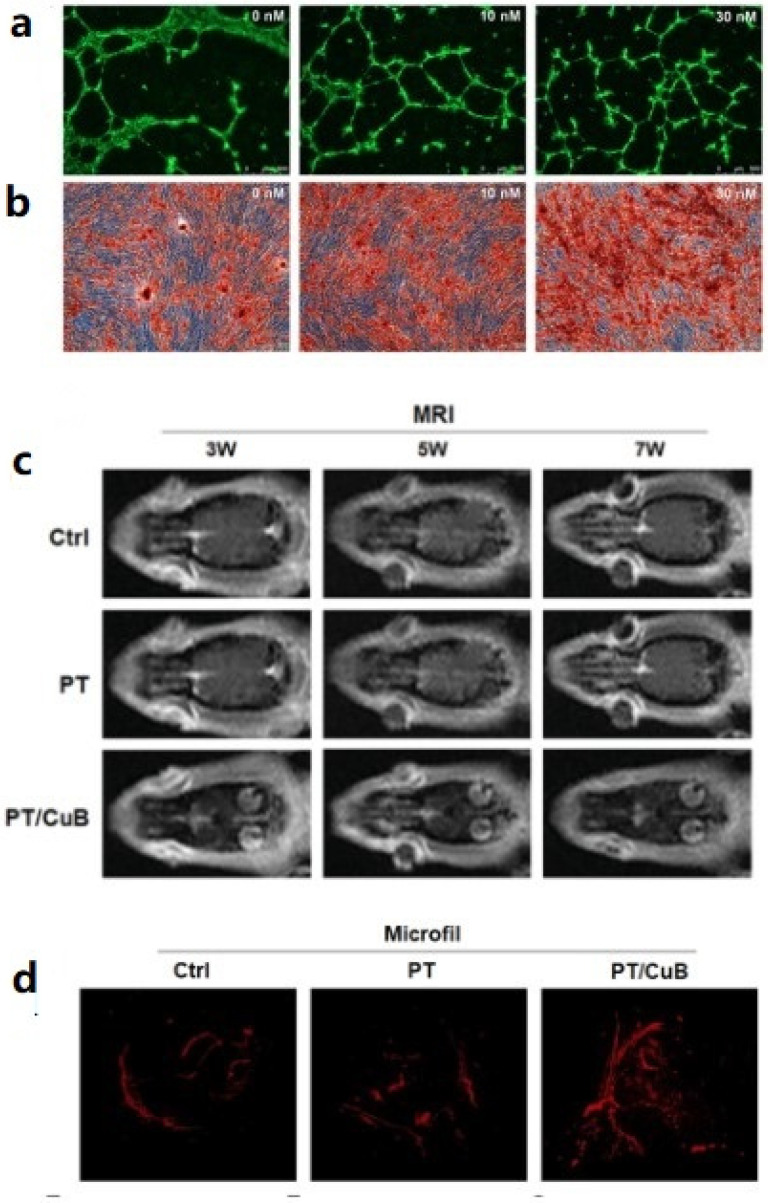
CuB has excellent performance of angiogenesis and osteogenesis. (**a**) In vitro angiogenesis, (**b**) alizarin red staining graph to evaluate osteogenic performance, (**c**) MRI scan image of animal model, and (**d**) micro-CT to evaluate in vivo vascularization capacity after 8 weeks, Reprinted with permission from ref. [50]. Copyright 2021 Elsevier.

**Figure 6 ijms-23-05831-f006:**
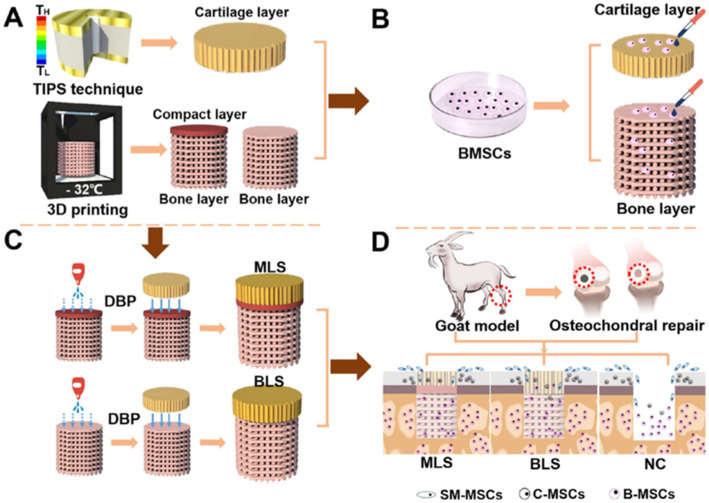
Schematic illustration of the MLS design, Reprinted with permission from ref. [54]. Copyright 2018 ACS Publications. (**A**,**C**) Preparation process of MLS, (**B**) Adding different functional structures of MSC, (**D**) Implantation of scaffolds in the defect of femoral cartilage in *goats*.

**Figure 7 ijms-23-05831-f007:**
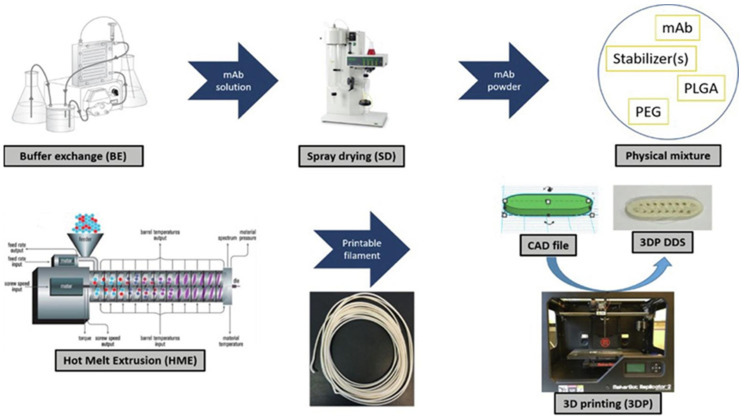
Preparation of PLGA-based Composite load mAb using FDM technology, Reprinted with permission from ref. [62]. Copyright 2021 IOP Publishing.

**Table 1 ijms-23-05831-t001:** Representative component regulation characteristics of PLGA-based scaffolds.

Composition Regulation	Description	Advantage	Limitations
Inorganic materials	β-TCP [33]	Mechanical strength	Degradation rate
	Mg [35]	Biological properties	Ion release
Growth factor	BMP-2 [40]	Biological properties	Scaffolds making restrictions
	VEGF [49]		Release rate
Drug	Salvianolic acid B [44]	Biological properties	Drug release rate
	Icariin [45]		
Cells	ADSC [31]	Biological properties	Operating environment
			Price control

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
