# Peer review of "Application of 3D-Printed, PLGA-Based Scaffolds in Bone Tissue Engineering"

_ijms, 2022, doi:10.3390/ijms23105831_

Round 1

Reviewer 1 Report

The manuscript reviews the research progress of 3D-printed PLGA-based scaffolds used in bone tissue engineering. The article is generally divided into 5 chapters, with deep and focused study of Physical and degradation behaviour of PLGA (1), Regulations of compositions in 3D-printed scaffolds (2), Structure design of 3D-printed scaffolds (3), Printing methods of 3D-printed scaffolds (4), Challenges and future perspective (5). The article has also introduction and conclusion sector.

During the analysis of the manuscript it was found out, that some actual and recent research, relevant in the study field is missing. It is recommended to add into review the following review research. Please explain how and in what way this article builds on and expands on the previous findings:

Gentile, P.; Chiono, V.; Carmagnola, I.; Hatton, P.V. An Overview of Poly(lactic-co-glycolic) Acid (PLGA)-Based Biomaterials for Bone Tissue Engineering. Int. J. Mol. Sci. 2014, 15, 3640-3659. https://doi.org/10.3390/ijms15033640 

Shue Jin, Xue Xia, Jinhui Huang, Chen Yuan, Yi Zuo, Yubao Li, Jidong Li,
Recent advances in PLGA-based biomaterials for bone tissue regeneration,
Acta Biomaterialia, Volume 127, 2021, Pages 56-79,

Please add as well information on some additional specific research from the study area - some recent research and some dated to the end of 20th century:

Wen Zhao, Jiaojiao Li, Kaixiang Jin, Wenlong Liu, Xuefeng Qiu, Chenrui Li,
Fabrication of functional PLGA-based electrospun scaffolds and their applications in biomedical engineering, Materials Science and Engineering: C,
Volume 59, 2016, Pages 1181-1194,

Sambit Sahoo, Siew Lok Toh, James C.H. Goh,A bFGF-releasing silk/PLGA-based biohybrid scaffold for ligament/tendon tissue engineering using mesenchymal progenitor cells,Biomaterials, Volume 31, Issue 11, 2010, Pages 2990-2998, ISSN 0142-9612,

Mouthuy, P.-A., El-Sherbini, Y., Cui, Z., and Ye, H. (2016) Layering PLGA-based electrospun membranes and cell sheets for engineering cartilage–bone transition. J Tissue Eng Regen Med, 10: E263E274. doi: 10.1002/term.1765. 

Sheikh, F. A., Ju, H. W., Moon, B. M., Lee, O. J., Kim, J.-H., Park, H. J., Kim, D. W., Kim, D.-K., Jang, J. E., Khang, G., and Park, C. H. (2016) Hybrid scaffolds based on PLGA and silk for bone tissue engineering. J Tissue Eng Regen Med, 10: 209221. doi: 10.1002/term.1989.

W. Schloegl, V. Marschall, M.Y. Witting, E. Volkmer, I. Drosse, U. Leicht, M. Schieker, M. Wiggenhorn, F. Schaubhut, S. Zahler, W. Friess,
Porosity and mechanically optimized PLGA based in situ hardening systems,
European Journal of Pharmaceutics and Biopharmaceutics,Volume 82, Issue 3, 2012, Pages 554-562, ISSN 0939-6411, https://doi.org/10.1016/j.ejpb.2012.08.006.

Hong, Y.J., Bae, S.E., Do, S.H. et al. Decellularized PLGA-based scaffolds and their osteogenic potential with bone marrow stromal cells. Macromol. Res. 19, 1090 (2011). https://doi.org/10.1007/s13233-011-1004-8

Duoyi Zhao, Tongtong Zhu, Jie Li, Liguo Cui, Zhiyu Zhang, Xiuli Zhuang, Jianxun Ding, Poly(lactic-co-glycolic acid)-based composite bone-substitute materials, Bioactive Materials, Volume 6, Issue 2, 2021,  Pages 346-360, ISSN 2452-199X, https://doi.org/10.1016/j.bioactmat.2020.08.016.

Author Response

We thank the reviewers for correcting the shortcomings in the manuscript. We have revised it accordingly in the manuscript.

Reviewer 2 Report

The paper presents an overview of the topic of modern developments in 3D printing of various PLGA-based composites for the development of biocompatible scaffolds and facilitating bone tissue regeneration.

The work itself is somewhat sloppy, which makes it difficult to understand the results. Most of the abbreviations are undisclosed and may be unfamiliar to the reader.

The review contains not too many references, which is probably due to the novelty of the subject.

The conclusions are very weak and should be reworked.

The manuscript can be published after the conclusions have been revised.

1) Line 52. PLGA definition should be given in the main text

2) Line 71. I like Daltons, but now g/mol should be used instead of following new SI rules.

3) Line 115. etc. TCP, PCL, PT, PTM, HA, BMP, OCD, and FDM abbreviations must be given in the text.

4) Line 135. a space symbol should be used between a number and definition. Also, MPa should be written instead of Mpa.

5) Line 158. bMP should be the BMP?

6) Line 315 So mAb or MAB?

Author Response

We thank the reviewers for correcting the shortcomings in the manuscript. We have revised it accordingly in the manuscript.

1)、3) The corresponding definitions have been added to the manuscript

2)、4) It has been revised accordingly in the manuscript

5)  Should be the BPM

6)  Should be the mAb